# Supportless Lattice Structure for Additive Manufacturing of Functional Products and the Evaluation of Its Mechanical Property at Variable Strain Rates

**DOI:** 10.3390/ma15227954

**Published:** 2022-11-10

**Authors:** Mayur Jiyalal Prajapati, Chinmai Bhat, Ajeet Kumar, Saurav Verma, Shang-Chih Lin, Jeng-Ywan Jeng

**Affiliations:** 1High Speed 3D Printing Research Center, National Taiwan University of Science and Technology, No. 43, Sec. 4, Keelung Rd., Taipei 106, Taiwan; 2Department of Mechanical Engineering, National Taiwan University of Science and Technology, No. 43, Sec. 4, Keelung Rd., Taipei 106, Taiwan; 3Department of Design, Indian Institute of Technology Guwahati, Guwahati 781039, India; 4Graduate Institute of Biomedical Engineering, National Taiwan University of Science and Technology, No. 43, Sec. 4, Keelung Rd., Taipei 106, Taiwan; 5Academy of Innovative Semiconductor and Sustainable Manufacturing, National Cheng Kung University, No. 1, University Rd., Tainan 701, Taiwan

**Keywords:** multi-material additive manufacturing, advanced product design, support free lattice structure, mass customization, direct digital manufacturing (DDM)

## Abstract

This study proposes an innovative design solution based on the design for additive manufacturing (DfAM) and post-process for manufacturing industrial-grade products by reducing additive manufacturing (AM) time and improving production agility. The design of the supportless open cell Sea Urchin lattice structure is analyzed using DfAM for material extrusion (MEX) process to print support free in any direction. The open cell is converted into a global closed cell to entrap secondary foam material. The lattice structure is 3D printed with Polyethylene terephthalate glycol (PETG) material and is filled with foam using the Hybrid MEX process. Foam-filling improves the lattice structure’s energy absorption and crash force efficiency when tested at different strain rates. An industrial case study demonstrates the importance and application of this lightweight and tough design to meet the challenging current and future mass customization market. A consumer-based industrial scenario is chosen wherein an innovative 3D-printed universal puck accommodates different shapes of products across the supply line. The pucks are prone to collisions on the supply line, generating shock loads and hazardous noise. The results show that support-free global closed-cell lattice structures filled with foam improve energy absorption at a high strain rate and enhance the functional requirement of noise reduction during the collision.

## 1. Introduction

Technological innovation is an economic function through which new technologies are introduced into product and process development with a cost-effective approach. It includes exploring new technological possibilities and organizing the financial and human resources needed to transform them into valuable products and processes by sustaining the requisite activities [1]. Innovation in the process results in the development of a product with enhanced performance to meet future consumer demands [2]. This is the main driving force for the growth of an industry. Hence, adopting new technologies to create products that cater to growing consumer demands of mass customization and personalization, meeting the production and supply chain agility requirement with reduced time and cost, is of prime importance.

Since inception, the first commercial use of additive manufacturing (AM) started in 1987 with the use of a stereolithography (SL) process developed by 3D Systems, followed by non-SL systems such as fused deposition modeling (FDM) from Stratasys and laminated object manufacturing (LOM) from Helisys in the year 1991 [3]. The layer-by-layer deposition process provides design freedom in fabricating complex geometries obtained using topology optimization or bio-inspired designs with minimum material wastage [4,5]. Topology optimization is one way to optimize material distribution in a given space to achieve lightweight, functional parts using advanced metal AM processes such as selective laser melting (SLM) or wire arc additive manufacturing (WAAM) [6,7,8,9]. Along with the fabrication of complex geometries, lightweighting can be achieved by varying the infill material inside a prototype or a fabricated part. The use of architectured materials, such as lattice structures, is one of the most effective ways to achieve a high strength-to-weight ratio [10]. Latticing is one of the most effective ways to achieve a high strength-to-weight ratio. It optimizes material usage, thereby reducing printing time and energy consumption in product manufacturing [11,12]. Along with this, lattice structures endow multi-functional properties such as high energy absorption, vibration and acoustic damping and excellent thermal and fluid flow properties [13,14,15,16,17,18].

One significant challenge that hinders lattice structure manufacturing is the need for support structures during the printing process. The complex geometries of the lattice structures involve features such as parallel ledges and overhangs that require supporting structures [19,20,21]. The fabrication and removal of the supporting structures significantly increase the printing and post-processing time [22,23,24]. The fabrication and removal of the supporting structures significantly increase the printing and post-processing time. Moreover, the removal of support structures can cause damage to the required component, rendering it unsuitable for the designed application. The post-processing time can be reduced by innovative product design involving the principles of design for AM and post-process (DfAM&PP) [25,26]. Thus, the use of lattice structures provides lightweight and toughness, but the necessity of supporting structures limits its capabilities to be used as an infill in part. This challenge necessitates designing supportless lattice structures with optimized mechanical properties. One such support-free lattice structure can be designed by taking inspiration from the sea urchin (SU) morphology.

The sea urchin lattice structure can be fabricated support-free using the material extrusion (MEX) process, one of the most popular AM processes for prototyping and composite manufacturing [22,27]. The design of the SU unit lattice is based on the design for the MEX process, according to which the lattice structure should avoid any steep overhangs (>45°) and parallel ledges to be support free. When printed with the MEX process, the lattice structures significantly reduce printing and post-processing time. Another advantage of the support-free SU unit cell is that it can be transformed into a closed cell structure by adding a thin/thick membrane across the unit cell openings (local-close cell structure) or across the complete lattice structure (global-close cell structure). Further, these closed cells can be filled in with secondary infill material to generate multi-functional properties such as improved toughness, vibration and noise damping. Adding functional properties with the inclusion of secondary material and a robust support-free lattice design is highly advantageous in manufacturing products suitable for a given application at a low cost. Along with excellent mechanical and functional properties, the support-free design reduces manufacturing costs and energy, increasing the process’s sustainability.

In the present age of industry 4.0, a dynamic production and supply line is a prerequisite for the mass customization and personalization of products. At present, AM is utilized to manufacture different modules of a product to reduce the number of parts required for assembly [28]. Other researches involve designing various gripping or holding devices and manufacturing them using AM technologies [29,30]. The future additive manufacturing production supply line should be able to accommodate a product with different shapes and sizes with minimum or no changes in tooling according to customer requirements. Looking at this scenario, the present study demonstrates an innovative 3D-printed universal puck design to accommodate different shapes and sizes of products in the supply line during the transfer from one station to another. The products in the present study are of different shapes and sizes based on functional and aesthetic requirements. Each of these can easily fit in a designed universal puck to meet the requirement of a future dynamic supply line. The proposed multipurpose puck design with additive manufacturing can accommodate multiple bottle design formats, enabling the same pucks to be used across all supply lines. The pucks are designed using the supportless SU lattice structure to achieve high strength with lightweight, which is required on the high-speed filling line. Since the SU lattice structure can be printed support free in any direction within the design domain, it can be easily accommodated in complex shapes without compromising the mechanical properties. High-speed travel can generate shocks and noise if a collision occurs between successive pucks. This challenge is countered by introducing polyurethane (PU) foam as a secondary material inside the closed cell cavity with good noise and shock absorption capabilities. Previous studies showed that the SU lattice structure can be printed supportless but were printed only in one orientation [22,27,31]. The present study also investigates the printing of supportless SU lattice structures in different orientations based on the design for additive manufacturing (DfAM) principles such that it can be easily accommodated in complex shapes without compromising mechanical properties. The application of PU Foam as secondary infill material in the close cell is meant for noise and shock absorption during high-speed travel on the supply line, unlike the previous study [26]. Since the collision speeds of the puck are unknown, variable strain rate compression testing was performed on unfilled and foam-filled global close lattice structures. At higher strain rates, the unfilled lattice structures disintegrate entirely, whereas the foam-filled lattice structures show progressive deformation with high energy absorption and crash force efficiency. The improved mechanical properties, with the addition of PU foam, make this design solution an appropriate choice for the universal puck design and can be applied in other complex designs required for industrial applications.

## 2. Materials and Methods

### 2.1. Design of Supportless Lattice Structure

The supportless lattice structures are based on the design for additive manufacturing (DfAM) principle for the MEX process. According to DfAM for MEX process, two critical features are to be evaluated before the design of a support-free lattice structure, (1) overhang angle and (2) bridging distance (Figure 1). Overhang is defined as the maximum angle up to which a feature can be 3D printed support free with optimized process parameters. A maximum overhang angle of 45° is considered a safe approximation. However, the angle varies significantly across different 3D printer brands, material used for 3D printing and also the required printing quality based on application [32]. A bridge is a horizontal surface supported by two or more features. The bridging distance depends on the melt viscosity of the extruded material and its cooling rate. A material with higher melt viscosity has a longer bridging length than a low viscous material. The bridging distance can be optimized by adjusting the printing temperature and the cooling speed [33,34]. The features were printed using the printing parameters set for PETG filament and are listed in Table 1. For overhangs, it was found that filament dropping starts at 60°, whereas, for bridge distance, it was observed that filament sagging increases after 5 mm distance. Thus, to design a supportless lattice structure using PETG material, an overhang of less than 60° and no bridging distance is required.

Based on the above criteria, the supportless Sea Urchin lattice structure is one of the few lattice structures which can be printed support free. The shape of the support-free unit cell is inspired by the sea urchin morphology. The unit cell is designed considering the overhang angle and bridging distance. Figure 2a shows the draft angle analysis, which shows that all the features of the single unit cell are well within 60° angle. The other parameter is bridge length which is mostly seen in truss-based lattice structure. The bridge can be formed during design of unit cell or during tessellation of unit cell.

A unit cells are arranged or tessellated into the 3D design space to obtain a complete lattice structure. Hence, the important consideration for support-free printing is to make sure that there are no bridges within unit cell or in tessellated lattice structure. Among different types of tessellations, periodic tessellation of unit cells is adopted as the design strategy in this study [35,36]. Using lattice structures for the lightweight design of a component using the MEX process is quite a challenging process. The support-free fabrication of lattice has to evaluate for all print directions to obtain final parts defect-free. In real-world application, as the parts are geometrically complex in shape, lattice direction can change the direction to accommodate the design space. Further, during AM process optimization, the 3D-printed parts can be printed in any angle as per the print time, and this also changes the direction of lattice.

Thus, it becomes necessary that the lattice structure be support-free when printed in different orientations. Figure 3 shows the draft analysis of a truss-based and a surface-based lattice structure. The draft angle is measured with respect to the vertical axis depicted by red arrows in Figure 3. BCC lattice structure shown in Figure 3a can be easily fabricated support free as all overhang angles are within 45° and no bridges are there. However, at 45° print orientation, as observed in Figure 3b, the BCC truss changes in horizontal and vertical beam forming bridge and thus requirement of support structure in the horizontal beam to avoid sagging during printing, compromising the structure’s overall mechanical strength. Other surface-based lattice structures, like the gyroid lattice structure, which have arch-like features, can be printed support free in 0° orientation as shown in Figure 3c. Still, as the orientation changes to 45°, the unsupported regions increase, making it difficult to print it support-free using the MEX process. The overhang defects for both, BCC and gyroid lattice structures are highlighted with yellow circles in Figure 3b,d.

Figure 4 shows that the sea urchin lattice structure can also be printed support free in 45° orientation. The red regions observed in the draft analysis of 45° orientation are well within the bridging distance; thus, it can be printed support free. This can be seen in the printed lattice structures in both orientations shown in Figure 4a,b. Such a unit cell design can be useful for manufacturing complex geometries with high strength and support-free, reducing manufacturing cost and time.

### 2.2. Materials

The closed cell material is printed with PETG filament (Day Tay Plastic Inc. Ltd., Taichung City, Taiwan). Dogbone specimens were printed according to ASTM D638-14 type ‘IV’ to estimate mechanical properties using tensile testing [37]. Young’s modulus of 2312 MPa was obtained with a fracture strength of 55.6 MPa. The secondary infill material used here is PU Foam (Smooth-On FlexFoam-iT!™ 15 Tuff Stuff, Phoenix, Arizona, U.S.A. Compression specimens with dimensions of 50 × 50 × 25 (L × B × H) were cut according to ISO 3386-1 using a band saw [38]. The elastic modulus obtained after compression testing was 1.084 MPa.

### 2.3. Support-Free Additive Manufacturing of Lattice Structure 

Global closed-cell lattice structures are fabricated using PETG filament using the primary nozzle. The design parameters of the global closed-cell lattice structure are listed in Table 1, and the printing parameters set for PETG material are listed in Table 2. PU foam is injected using the secondary foaming system (Figure 5). To inject PU foam, the G-code is modified to command the secondary foaming system to arrive at the filling position, initiate the mixing of polyol and isocyanate and inject it inside the lattice structure. The foam is allowed to expand and fill the internal cavity of the lattice structure [26].

### 2.4. Compression Test

Uniaxial compression testing was performed for the three specimens of each design using an MTS Insight Universal Testing Machine (MTS System Corporation, Cary, NC USA) at room temperature. The compression test was performed for 50% of the deformation of the initial height at three different crosshead speeds of 5 mm/min, 50 mm/min and 100 mm/min. The corresponding strain rates are 0.125 s^−1^, 1.25 s^−1^ and 2.5 s^−1,^ which are calculated using Equation (1).
(1)$$\mathrm{strain}\;\mathrm{rate}\;\left(s^{-1}\right)=\frac{\mathrm{crosshead}\;\mathrm{speed}}{\mathrm{sample}\;\mathrm{size}}$$

The loading direction was kept at 0° to the print direction to assess the compressive strength of the lattices. Load vs. deformation curves were obtained and analyzed to obtain energy absorption and crash force efficiency up to 50% compression of the lattices. 

## 3. Results and Discussion

### 3.1. Variable Strain Rate Compression Testing 

The load vs. deformation data obtained from quasi-static compression testing at different strain rates were analyzed to evaluate the mechanical properties of the unfilled and foam-filled lattice structures, as seen in Figure 6. Solid lines represent the deformation behavior of unfilled structures, while dotted lines represent the deformation behavior of foam-filled lattice structures. The deformation pattern of each design at various strain rates is shown in Figure 7. It can be seen from Figure 7a that at a lower strain rate of 0.125 s^−1^, the unfilled lattice structure deforms with sudden fractures at multiple locations after the initial plastic deformation. This is also seen in the load vs. deformation trend marked by a sudden drop in load with deformation in the plateau region. As observed normally at a low strain rate, even though fractures occur soon after the elastic region, the structure fails layer by layer, increasing and decreasing the loading trend until the densification region. The same deformation behavior can also be observed in the foam-filled lattice structure (Figure 7b) at 0.125 s^−1^ but without abrupt fracture, as seen in Figure 7b and in the load deformation graph. This means that the secondary material, i.e., PU foam adds toughness to the lattice structure and makes it more resistant to failure during high load acting on it.

Now, as the strain rate increases, the deformation and fracture pattern in the unfilled lattice structure suddenly changes during loading. The severity of fracture increases with an increase in the strain rate, as seen in Figure 7c,e. The nature of failure changes from elastic–plastic to a more brittle nature, and this can be observed in Figure 6. The abrupt brittle failure at the onset of the plateau region leads to a complete failure of structure which makes it unfit for energy absorption applications. Hence, at a high strain rate, the unfilled lattice structure changes its failure pattern along with material behavior. This behavior is undesirable for manufacturing functional parts intended for critical application in the industry and other sectors where the strain rate cannot be controlled. Whereas filling a secondary material foam inside the lattice structure can help to counter this phenomenon. As seen in Figure 6, at a high strain rate of 1.25 s^−1^ and 2.5 s^−1^, the foam-filled lattice structure shows elastic plastic failure rather than brittle failure shown by its counterpart. Although at a high strain rate, the load suddenly drops after the end of the elastic region but it soon recovers, and this phenomenon continues until densification is reached. As observed in Figure 7d,f, the foam-filled lattice structure is able to maintain structural integrity, the same as seen in the low strain rate of 0.125 s^−1^. At a very high strain rate of 2.5 s^−1^, as observed in Figure 7f, the structure disintegrates only during the densification region, whereas the unfilled lattice structure completely disintegrates soon after the elastic region. The deformation pattern, too, is almost similar irrespective of strain rate due to foam inside the structure. The foam provides cushioning support, thereby increasing the toughness and energy absorption. 

To evaluate the improvement in mechanical properties quantitatively, crash force analysis was performed for unfilled and foam-filled lattice structures. Crash force analysis was performed by comparing the specific energy absorption (SEA) and crash force efficiency (CFE) of both designs [39,40,41]. SEA is defined as energy absorption (EA) per unit mass and is calculated as
(2)$$\mathrm{SEA}=\frac{\mathrm{EA}}{m}$$

Here, m is the weight of the specimen. Energy absorption can be calculated by integrating the area under the load vs. deformation curve. It is expressed as
(3)$$\mathrm{EA}=\int_{0}^{d}x.F\left(x\right).\mathrm{dx}$$
where F is the crushing force, and x is displacement. Higher SEA shows better crashworthiness of the lattice structure. To evaluate the crash force efficiency, it is necessary to calculate mean force (P_mean_) and peak force (P_max_). P_max_ can be determined from the maxima of the load vs. deformation, whereas, P_mean_ is defined as
(4)$$P_{\mathrm{mean}}=\frac{\mathrm{EA}}{d}$$

Crash force efficiency can be calculated as
(5)$$\mathrm{CFE}\left(\%\right)=\frac{P_{\mathrm{mean}}}{P_{\max}}\times 100$$

CFE is a metric used to define the efficiency of energy absorption of the lattice structure. Higher CFE indicates that the structure is more stable during the deformation. 

Using Equations (2)–(5), mechanical properties were evaluated and are listed in Table 3. It can be seen that the energy absorption for the unfilled lattice structure decreases drastically from 42.15 J to 9.015 J as the strain rate increases from 0.125 s^−1^ to 1.25 s^−1^. Energy absorption for 2.5 s^−1^ is similar to that of 1.25 s^−1^. This sudden decrease in energy absorption can be attributed to the change in the deformation behavior from elastic–plastic to brittle failure. In the case of foam-filled lattice structures, the energy absorption is reported to be 54.42 J, 55.85 J and 57.12 J for strain rates of 0.125 s^−1^, 1.25 s^−1^ and 2.5 s^−1^, respectively. This shows that irrespective of the strain rates, the energy absorption is similar for the foam-filled lattice structures. Foam provides the necessary toughness and restricts brittle failure in the structural material. This contributes to the increase in energy absorption compared to the unfilled lattice structures at higher strain rates. 

Further, it can also be noticed that the P_mean_ is higher for foam-filled lattice structures when compared to unfilled lattice structures at higher strain rates. This is intuitive since there is a significant increase in EA for the foam-filled lattice structures at higher strain rates. P_max_ and P_mean_ are used to calculate CFE. It can be seen from Table 3 that the crash force efficiency of the unfilled lattice structure decreases drastically as the strain rate increases. This means that if the crushing force is applied at a higher strain rate, the stability of the unfilled lattice structure will reduce significantly. Whereas, for foam-filled lattice structures, the structural integrity is still maintained at higher strain rates making it more stable for higher strain rate applications.

Thus, it can be seen that by adding a secondary material, such as PU foam, the energy absorption characteristics of the lattice structure can be improved significantly with a marginal increase in weight. Such properties are a prerequisite to manufacturing many industrial-grade components that require high toughness and lightweight structure. By using this innovative DDM strategy with a cost-effective AM technology such as the MEX process, manufacturing time and costs can be significantly reduced to generate more profit.

### 3.2. Case Study

This case study highlights the importance and application of the above-mentioned study for industrial goods manufacturing. A universal puck is designed using the supportless SU closed-cell lattice structures filled with PU foam. The product is also designed based on DfAM&PP principles for the MEX process. Pucks form an integral part of consumer-based industries that are required to firmly hold the products across the processing line. Conventionally, variable products have different sizes and shapes and require different pucks for holding and transfer in the conveyor line or during packing. This induces the processing, laboring and handling constraints impacting the overall outcome of the industries [42]. Apart from that, the production of individual pucks through injection molding involves designing different molds, which increases industries’ handling costs [43,44].

The complexity will further increase as the industry moves from mass manufacturing to mass customization and personalization of products in the era of AM and industry 4.0. Mass customization can be explained as the production of highly variant products for the user at a cost close to mass production. This brings a new challenge for the industry to make the fabrication process flexible, robust and agile. Additive manufacturing and DfAM can be utilized to design innovative solutions which otherwise was very challenging with traditional manufacturing. In this case study, a universal puck is designed to accommodate various designs of products. Here, the products are cosmetic bottles of different shapes and sizes that are required to be moved to the filling station and further to the packaging across the high-speed supply line, as seen in Figure 8. 

The whole case study is divided into three stages: (1) Designing of the universal puck, (2) Additive manufacturing of the designed puck and bottles using the material extrusion process and visual inspection of printing details and gripping, (3) evaluation of the performance of different pucks based on the fabrication and industrial criteria. 

For stage 1 of the study (design of the universal puck), the five most common bottle designs used in various consumer products, such as shampoos, conditioners and perfumes, are considered. The dimensional details of these bottles are shown in Figure 8a–e. The hypothetical fluid of density equivalent to glycerin (1.26 g/cm^3^) is used as the filler material in each bottle, as shown in Figure 8a–e. The reason behind using glycerin as the filler material is that it is the precursor for many consumer-based products. Each bottle’s Centre of Mass (COM) dictates the dimension of the universal puck design. The lower the COM, the more stable the bottles will be, but since most of the consumer-based products are slender, their stability is less. The slots for each bottle are carved in the universal puck to accommodate the COM within the puck. This would prevent the bottles from wobbling in the processing line. Figure 8f shows the cross-section of the designed puck with slots carved for different bottles. Table 4 compares the height of COM of each bottle and the accommodating height for that bottle in the puck.

In this study, two universal pucks were additively manufactured. The first puck was manufactured with supportless SU lattice infill, as shown in Figure 9a. The second puck was manufactured with supportless SU lattice infill and was filled with PU foam inside the cavities, as shown in Figure 9b. The foam as the secondary material was believed to increase strength and damage resistance, thereby increasing usability. Moreover, the foaming material is also believed to be noise-absorbent [45]. The foam as the secondary material was believed to increase strength and damage resistance, thereby increasing usability. Moreover, the foaming material is also believed to be noise-absorbent [46]. The performance of both pucks was evaluated based on the printing and post-processing time criteria.

Along with the fabrication criteria, the strength of these pucks was evaluated to mimic the actual condition of collision in the processing line of the industries. Moreover, the noise reduction of these pucks upon collision is an essential industrial criterion. Using these pucks in industries would only be justified if the collision noise level is less than 85–90 dBA, as prolonged exposure to noise greater than 90 dBA would cause hearing-related problems in workers [46].

Figure 10 represents the second stage of the study, where the top view of the printed universal puck is shown. It can be observed in Figure 10 that each bottle can be accommodated in its respective slots. Considering the bottle designs, +0.4 mm tolerance is given to the puck while printing to facilitate easy picking of the bottle from the puck in the processing line. Figure 11 shows all printed bottles that are fitted in the universal puck without any risk of wobbling or damage.

The performance of the universal pucks is evaluated in the third stage of the study. Table 5 lists the performance of both pucks under different evaluation criteria. Considering the printing and post-processing time, the empty puck has an upper edge with a printing time of 353 min compared to a printing time of 362 min for a foam-filled puck. The excessive time of 9 min for the foam-filled puck was due to the simultaneous foam-filling process using the second nozzle and the foaming reaction. However, the mechanical properties evaluation listed in Table 2 and Shown in Figure 10a reveals that the initial stiffness of the puck increased by 133% upon the use of secondary foam. Moreover, the foam-filled puck could bear the peak compressive load of 3439 N, which is approximately three times the load that an empty puck can bear. Apart from load-bearing abilities, the energy absorption of a foam-filled puck was 64.76 J, which is 126% more than an empty puck’s energy absorption. The substantial increment in the load-bearing and energy absorption abilities of the foam-filled puck makes them more resilient to damage, increasing their usability. Apart from the mechanical performance, the collision noise of the two pucks was evaluated by sliding one puck from a distance of 500 mm and elevation of 200 mm, as shown in Figure 12b,c. The noise generated upon the collision of two pucks was recorded using the noise-detecting microphone shown in Figure 10b,c. This experiment mimics the termination of the processing line in industries where several pucks collide at the end. The continuous collision generates an unpleasant noise which should be within the safe limit of 90 dBA. The collision experiment is repeated 20 times, and the average noise generated is calculated. The observation reveals that the noise level was reduced from 99.1 dBA to 82.7 dBA upon foam filling. The 16.5% reduction in the noise level makes the noise level within the industrial permitted limits. 

The case study successfully proves that the additively manufactured universal pucks with supportless SU lattice infill and secondary foam filling would be effectively utilized in the current and future product development. The pucks can be modified and manufactured easily with the changing bottle designs, reducing the overhead charges involved with injection molding. Apart from that, the secondary foam-filling also enhances the functional property by reducing the collision noise level, making the industrial environment more worker-friendly. 

## 4. Conclusions

In this study, the importance and application of the supportless SU lattice structure are highlighted via an industrial case study. Based on the DfAM for the MEX process, it was shown that the supportless SU lattice structure can be printed in different orientations. This makes it a versatile structure, fit to accommodate complex shapes without compromising the mechanical properties. PU foam is filled in the global closed SU lattice structure, and it was found that the mechanical properties improved when compared with the unfilled lattice structure at variable strain rates. Foam-filled SU lattice structures were then incorporated in a universal puck which was designed and 3D printed to accommodate several products (bottles) on the high-speed filling. Important conclusions made in the study are as follows:Variable strain rate analysis showed that failure of unfilled lattice structures changes from elastic–plastic to brittle with an increase in strain rate. The addition of PU foam improves the resilience of the lattice structure and provides the necessary toughness at higher strain rates.At the strain rate of 0.125 s^−1^ EA increases from 42.15 J to 54.42 J with the addition of foam, whereas at higher strain rates of 1.25 s^−1^ and 2.5 s^−1^, EA increases from 9.015 J to 55.85 J and 9.851 J to 57.12 J, respectively.Consequently, CFE at 0.125 s^−1^ increases from 65.41% to 77.85% with foam-filling, whereas at higher strain rates of 1.25 s^−1^ and 2.5 s^−1^, CFE increases from 12.72% to 75.1% and 13.5% to 75.7%, respectively.Visual inspection showed that all the bottles successfully fit in the designed universal puck without any wobble or dropping of the bottles.Mechanical and noise absorption testing of the universal pucks showed that foam-filled pucks have a 133% increase in stiffness, 126% increase in energy absorption and 16.5% noise reduction than unfilled pucks.

## Figures and Tables

**Figure 1 materials-15-07954-f001:**
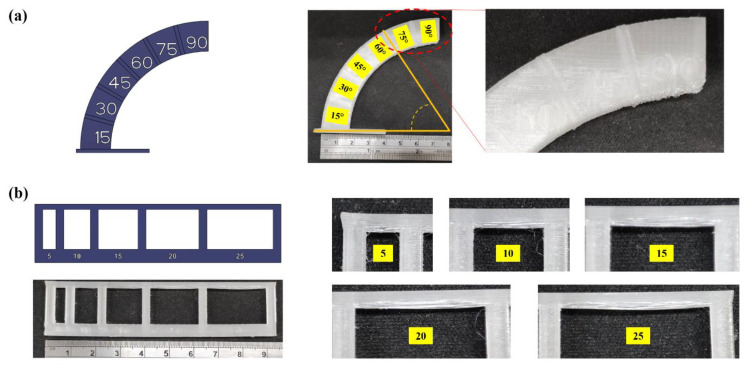
(**a**) Overhang angle and (**b**) Bridge distance for PETG material.

**Figure 2 materials-15-07954-f002:**
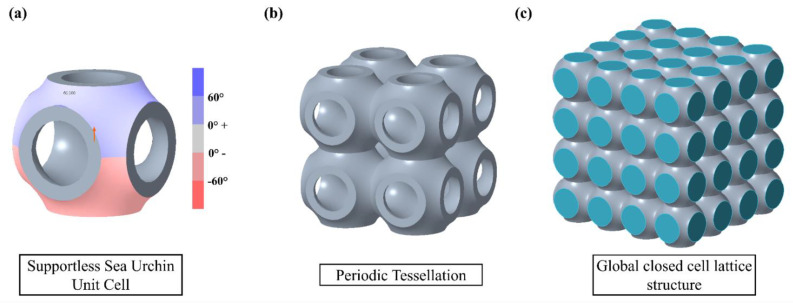
(**a**) Draft analysis of supportless SU unit cell design. (**b**) Periodic tessellation of open cells. (**c**) Global closed cell lattice structure.

**Figure 3 materials-15-07954-f003:**
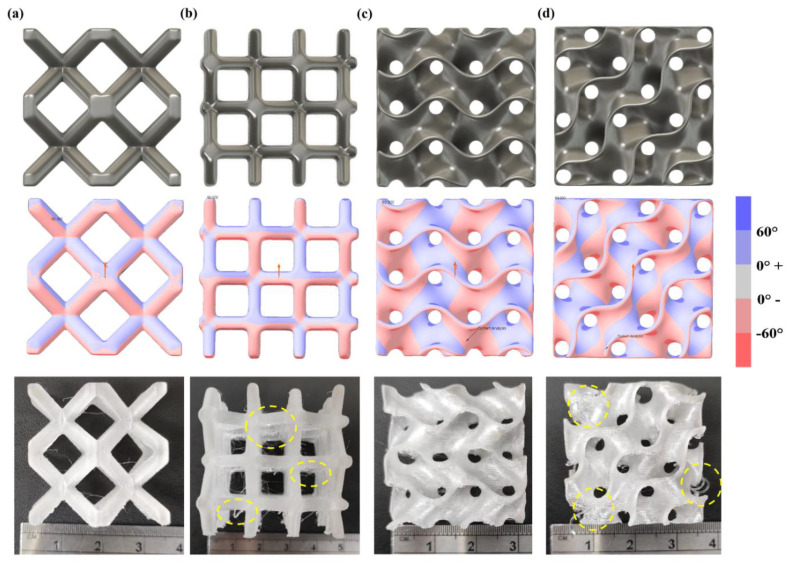
CAD design, draft analysis and additive manufacturing of BCC lattice structure in (**a**) 0° orientation and (**b**) 45° orientation and Gyroid lattice structure in (**c**) 0° orientation and (**d**) 45° orientation.

**Figure 4 materials-15-07954-f004:**
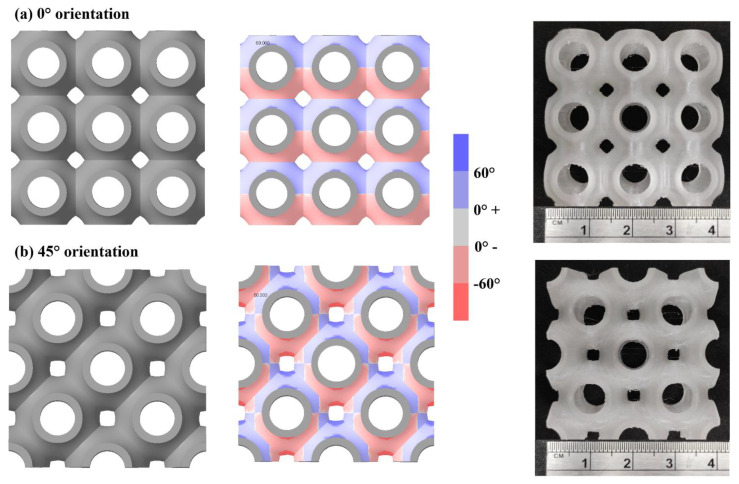
CAD design, draft analysis and additive manufacturing of SU lattice structure in (**a**) 0° orientation and (**b**) 45° orientation.

**Figure 5 materials-15-07954-f005:**
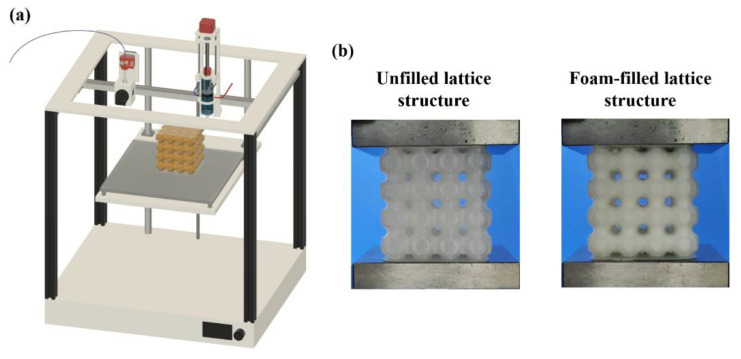
(**a**) CAD model of Hybrid FFF system, (**b**) 3D-printed unfilled and foam-filled lattice structures.

**Figure 6 materials-15-07954-f006:**
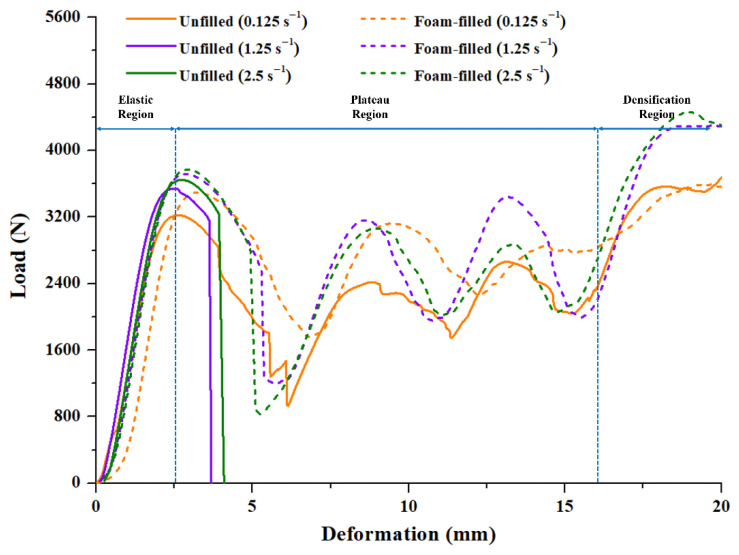
Load vs. deformation response unfilled and foam-filled lattice structures and 0.125 s^−1^, 1.25 s^−1^ and 2.5 s^−1^.

**Figure 7 materials-15-07954-f007:**
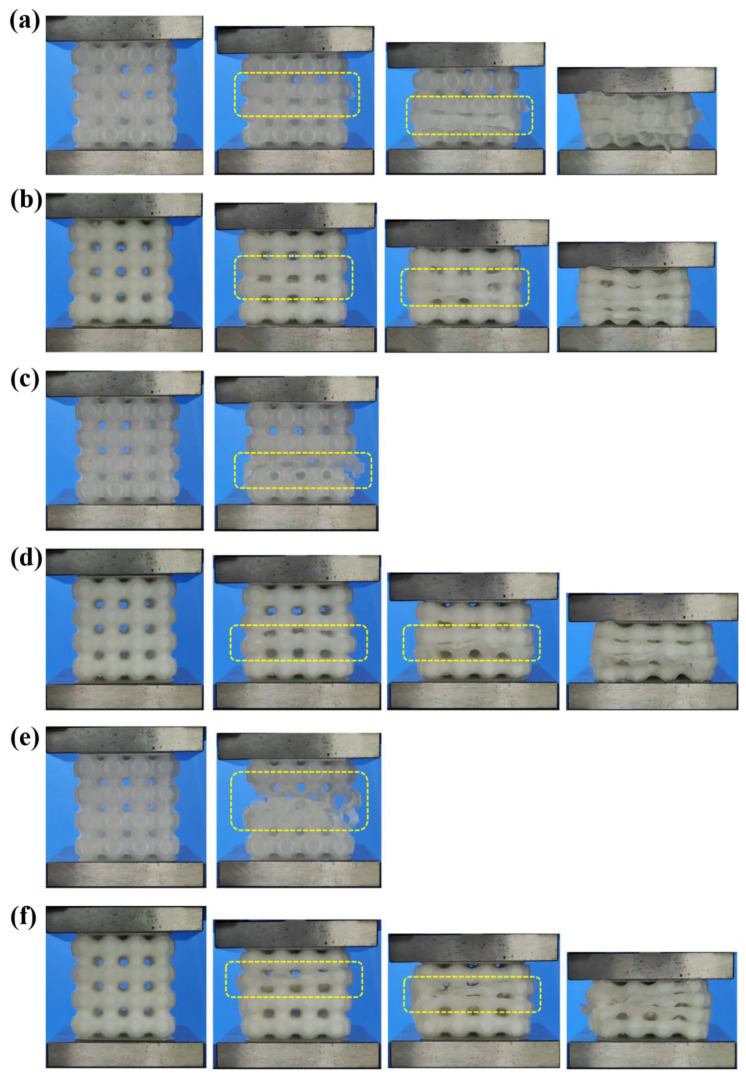
Deformation pattern of unfilled and foam-filled lattice structures at (**a**,**b**) 0.125 s^−1^, (**c**,**d**) 1.25 s^−1^, (**e**,**f**) 2.5 s^−1^.

**Figure 8 materials-15-07954-f008:**
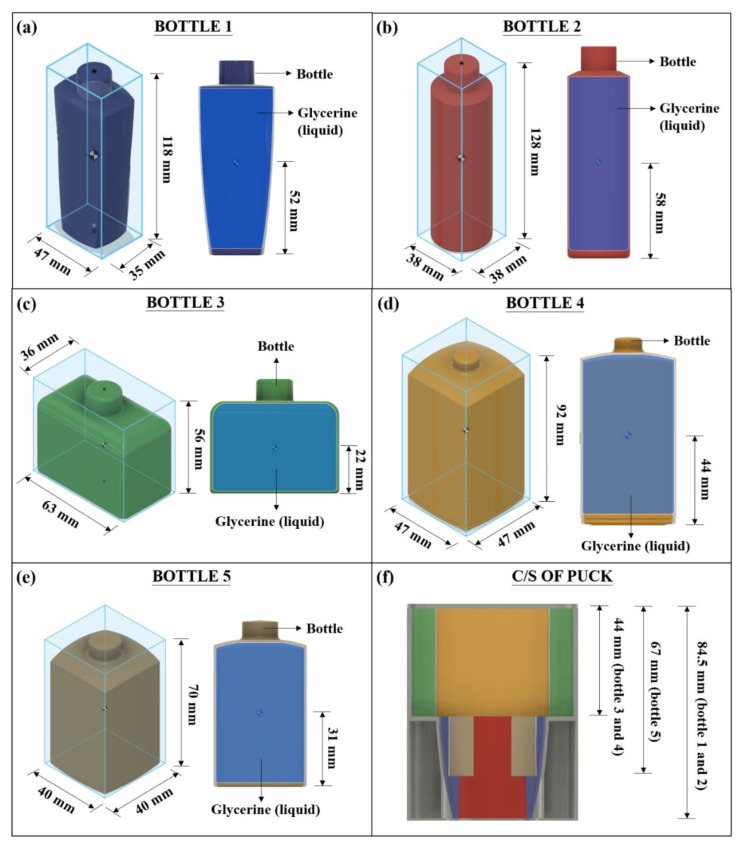
Dimensional details of designed bottles and their Center of Mass (COM) (**a**–**e**) and cross-section of the universal puck carved to hold different bottles (**f**).

**Figure 9 materials-15-07954-f009:**
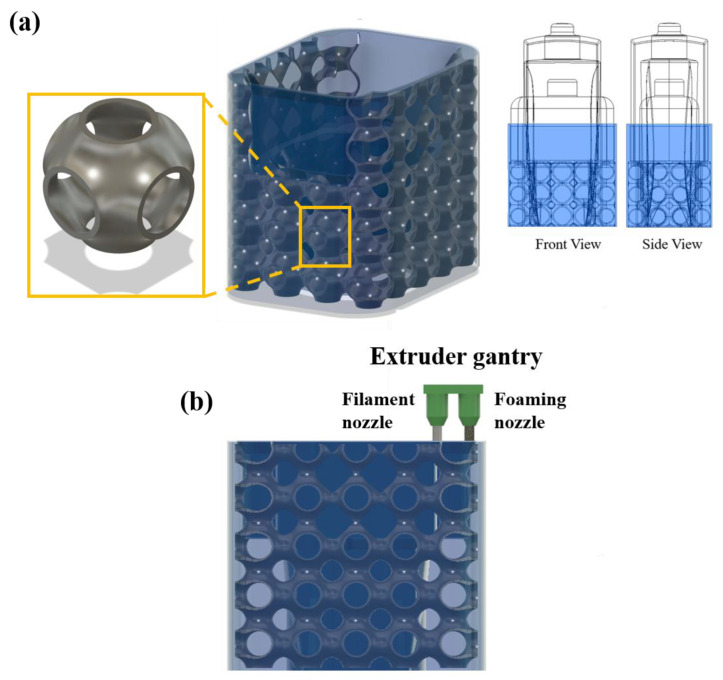
(**a**) Universal puck design with supportless SU lattice infill (**b**) Foam filling in the second puck using the second FDM nozzle.

**Figure 10 materials-15-07954-f010:**
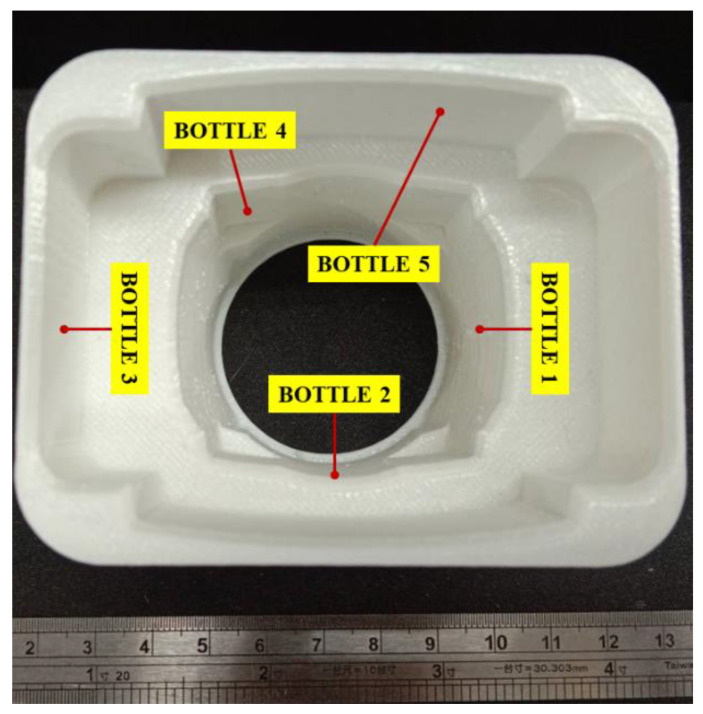
Top view of the printed universal puck with slots for five bottles.

**Figure 11 materials-15-07954-f011:**
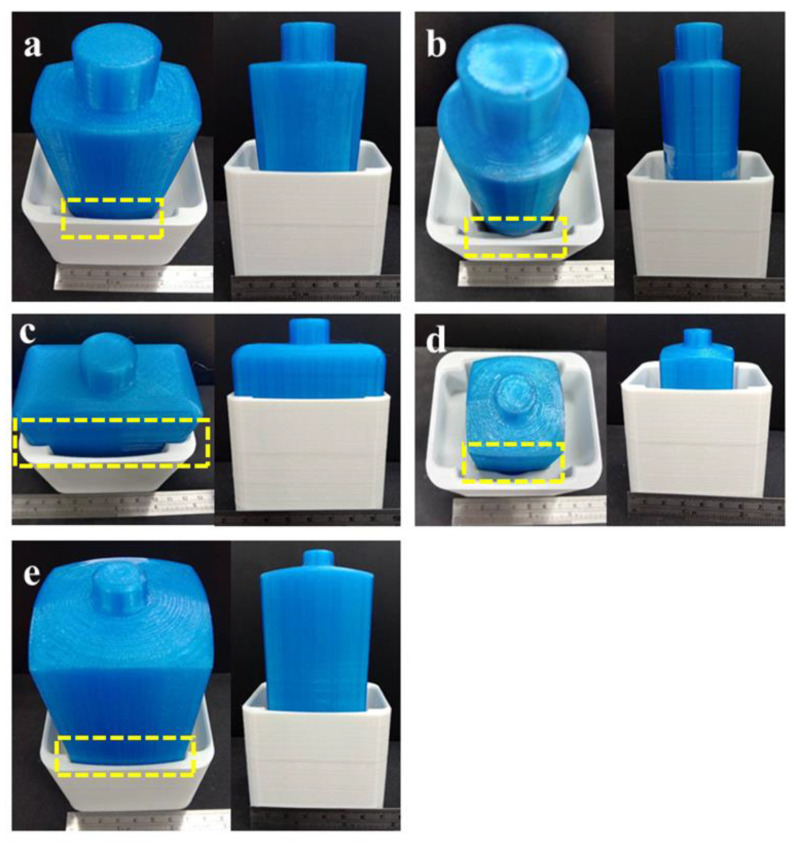
3D-printed bottles being placed in the universal puck (**a**) bottle 1 (**b**) bottle 2 (**c**) bottle 3 (**d**) bottle 4 (**e**) bottle 5.

**Figure 12 materials-15-07954-f012:**
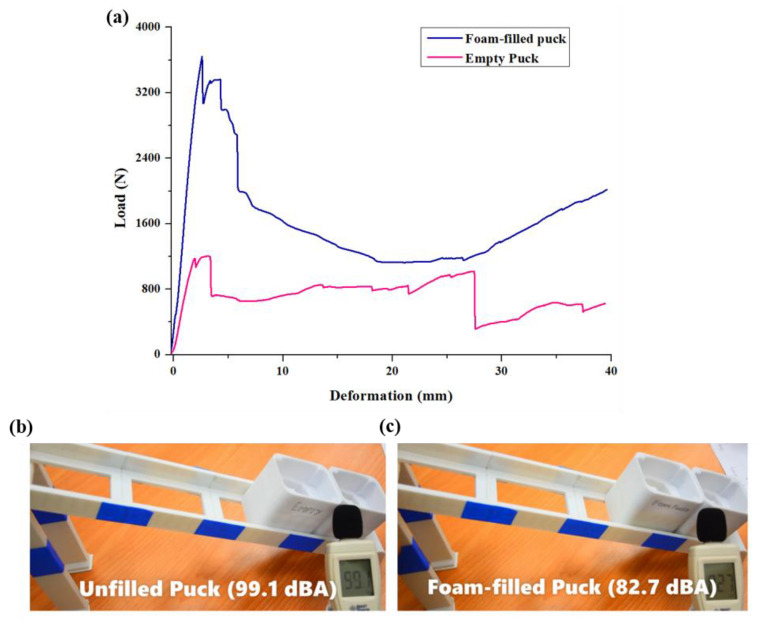
(**a**) Load vs deformation graph for unfilled and foam-filled puck, noise emission on collision between (**b**) unfilled puck and (**c**) foam-filled puck.

**Table 1 materials-15-07954-t001:** Design parameters for lattice structures.

Primary Cell Structure	Primary Material	Relative Density (RD)	Shell Thickness ‘T_s_’ = (r_1_ − r_2_) (mm)	Membrane Thickness ‘*M_t_*’ (mm)	Secondary Filled Material
Global closed sea urchin lattice structure	PETG	19%	0.5	0.5	PU Foam

**Table 2 materials-15-07954-t002:** Printing parameters for PETG filament.

Nozzle Diameter (mm)	Printing Temperature (°C)	Bed Temperature (°C)	Layer Height (mm)	Print Infill (%)	Printing Speed (mm/s)
0.4	220	60	0.2	100	90

**Table 3 materials-15-07954-t003:** Crash force analysis of unfilled and form-filled lattice structure.

Strain Rate (s^−1^)	Energy Absorption (J)		Specific Energy Absorption (J/g)		Peak Force (kN)		Mean Force (kN)		Crash Force Efficiency	
	Unfilled	Foam-Filled	Unfilled	Foam-Filled	Unfilled	Foam-Filled	Unfilled	Foam-Filled	Unfilled	Foam-Filled
0.125	42.15	54.42	2.4	2.62	3.22	3.49	2.11	2.72	65.41	77.85
1.25	9.015	55.85	0.52	2.68	3.54	3.72	0.45	2.8	12.72	75.1
2.5	9.851	57.12	0.56	2.75	3.65	3.77	0.49	2.86	13.5	75.7

**Table 4 materials-15-07954-t004:** Comparison of the center of mass (COM) of each bottle and their accommodating height in the universal puck.

Bottle Number	Center of Mass (mm)	Accommodating Height in Puck (mm)	Difference (mm)
1	52	84.5	32.5
2	58	84.5	26.5
3	22	44	22
4	44	44	0
5	31	67	36

**Table 5 materials-15-07954-t005:** Fabrication and mechanical properties comparison of the empty and foam-filled universal puck.

Universal Puck	Printing + Post-Processing Time (min)	Initial Stiffness (N/mm)	Peak Load (N)	Energy Absorption (J)	Collision Noise (dBA)
Empty puck	353	700	1197	28.56	99.1
Foam filled	362	1631	3439	64.76	82.7

## Data Availability

All the data is available within the manuscript.

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
