# Peer review of "Supportless Lattice Structure for Additive Manufacturing of Functional Products and the Evaluation of Its Mechanical Property at Variable Strain Rates"

_materials, 2022, doi:10.3390/ma15227954_

Round 1
Reviewer 1 Report
The manuscript entitled “Supportless lattice structure for additive manufacturing of functional products and the evaluation of its mechanical property at variable strain rates” is well-written and presented. Experiments on supportless lattice structures are conducted in detail. The writing is acceptable. Therefore, the authors recommend the minor revision. The comments are:
1. In Introduction, references are required to prove the statements “The fabrication and removal of the supporting structures significantly increase the printing and post-processing time.”
2. The legends in Figures 2 and 3 are not very clear. Please improve them.
3. In Figure 5, the legends can be put inside the figure.
4. “Apart from the mechanical performance, the collision noise of the two pucks 379
was evaluated by sliding one puck from a distance of 500 mm and elevation of 200 mm, 380
as shown in Figures 9b and 9c.”
Figure 9 does not have the subfigures.
5. In the caption of Figure 11, where is c)?
Author Response
The responses have been submitted in a word document format. Please see the attachment.

Reviewer 2 Report
Interesting work. Hereafter several comments can be found:
1) The state of the art should be enhanced with works like
(a) for methodology: Kladovasilakis, N., Tsongas, K., Karalekas, D., & Tzetzis, D. (2022). Architected Materials for Additive Manufacturing: A Comprehensive Review. Materials, 15(17), 5919.
(b) for applications: Papacharalampopoulos, A., Karapiperi, A., & Stavropoulos, P. (2021). Humanitarian engineering design methodology for AM metallic products: A smart mobility platform case. Procedia Cirp, 97, 59-65.
2) The added value of the work should be explicitly stated especially with respect to similar works like [18-20] therein
3) the foam depiction should be enhanced in a single photo, indicating its differentiation from the rest of the material on the part
4) the structure needs to be changed a bit; bottles are presented out kind of out of the blue
5) the first paragraph of the conclusions is almost trivial; maybe the focus should change towards a more concrete conclusion
Author Response

(The authors gave the same response as above.)

Reviewer 3 Report
The paper entitled "Supportless lattice structure for additive manufacturing of functional products and the evaluation of its mechanical property at variable strain rates" proposes an innovative design solution based on the design for additive manufacturing (DfAM) and post-process for manufacturing industrial-grade products by reducing AM time and improving production agility. From my point of view, the topic is of great interest, some comments:
· A graphical abstract would add interest to catch the eye
· Please put in the abstract some quantifiable result that adds visibility to the paper.
· Please add a picture of the experimental equipment could give visual information on the equipment.
· Add scale label on Figure 3 and Figure 4 colorbars.
· Add recent contributions that would be interesting to discuss is that of topological optimization. In the case of WAAM (metal material) is not the most suitable, although, there are several paper dealing with this topic:
o https://doi.org/10.1089/3dp.2021.0008 o https://doi.org/10.1016/j.addma.2019.06.010 o https://doi.org/10.3390/polym14112177· The conclusions could be enriched and presented in bullet format.
These comments are intended to improve a correct experimental works done in 3D printing of case scenario parts
Author Response

(The authors gave the same response as above.)

Round 2
Reviewer 2 Report
It seems that the comments have been addressed